# Spectrum of Genetic Mutations in Korean Pediatric Acute Lymphoblastic Leukemia

**DOI:** 10.3390/jcm11216298

**Published:** 2022-10-26

**Authors:** Jae Won Yoo, Ari Ahn, Jong-Mi Lee, Suejung Jo, Seongkoo Kim, Jae Wook Lee, Bin Cho, Yonggoo Kim, Myungshin Kim, Nack-Gyun Chung

**Affiliations:** 1Department of Pediatrics, Seoul St. Mary’s Hospital, College of Medicine, The Catholic University of Korea, Seoul 06591, Korea; 2Catholic Genetic Laboratory Center, Seoul St. Mary’s Hospital, College of Medicine, The Catholic University of Korea, Seoul 06591, Korea; 3Department of Laboratory Medicine, College of Medicine, The Catholic University of Korea, Seoul 06591, Korea

**Keywords:** mutation landscape, next-generation sequencing, acute lymphoblastic leukemia, pediatrics

## Abstract

The wide application of next-generation sequencing (NGS) technologies has led to the discovery of multiple genetic alterations in pediatric acute lymphoblastic leukemia (ALL). In this work, we aimed to investigate the mutational spectrum in pediatric ALL. We employed a St. Mary’s customized NGS panel comprising 67 leukemia-related genes. Samples were collected from 139 pediatric ALL patients. Eighty-five patients (61.2%) harbored at least one mutation. In B-cell ALL, the RAS pathway is the most involved pathway, and the three most frequently mutated genes were *NRAS* (22.4%), *KRAS* (19.6%), and *PTPN11* (8.4%). *NRAS* and *PTPN11* were significantly associated with a high hyperdiploidy karyotype (*p* = 0.018 and *p* < 0.001, respectively). In T-cell ALL, the three most frequently mutated genes were *NOTCH1* (37.5%), *FBXW7* (16.6%), and *PTEN* (6.2%). Several pairs of co-occurring mutations were found: *NRAS* with *SETD*, *NRAS* with *PTPN11* in B-cell ALL (*p* = 0.024 and *p* = 0.020, respectively), and *NOTCH1* with *FBXW7* in T-cell ALL (*p* < 0.001). The most frequent newly emerged mutation in relapsed ALL was *NT5C2*. We procured comprehensive genetic information regarding Korean pediatric ALL using NGS technology. Our findings strengthen the current knowledge of recurrent somatic mutations in pediatric ALL.

## 1. Introduction

The cure rate of childhood acute lymphoblastic leukemia (ALL) has significantly improved owing to contemporary risk-directed treatment, demonstrating a 5-year overall survival (OS) of over 90% in many clinical trials [1,2,3,4]. However, current therapies for ALL are associated with substantial toxic effects, and up to 20% of patients still show relapse after the initial therapy [5]. To further improve outcomes in the upcoming decade, trials need to focus not only on the molecular and immune therapies that might replace toxic chemotherapy but also on the biologic basis of the drug resistance that leads to relapse. In this regard, enhanced knowledge about the genetic background of childhood ALL is gaining importance.

The classification of B-cell precursor (BCP)-ALL and T-cell ALL (T-ALL) has been refined based on gene expression-based subgroups, and recent studies have further identified several novel rearrangements including *DUX4*-rearranged [6], iAMP21 and *MEF2D*-rearranged [7,8], *ZNF384*-rearranged [9], and *ETV6-RUNX1*-like B-ALL [8]. In addition, the wide application of next-generation sequencing (NGS) technologies has revealed more abnormal molecules and improved the understanding of ALL biology; the pathway most commonly altered in ALL is the transcriptional regulation of lymphoid development (*PAX5*, *IKZF1*, and *EBF1*) [10], and additional mutated pathways include tumor suppression and cell-cycle regulation (*TP53*, *RB1*, *PTEN*, and *CDKN2A*), RAS signaling (*NRAS*, *KRAS*, and *PTPN11*) [11], and epigenetic modification (*CREBBP*, *EP300*, *SETD2*, and *NSD2*) [12]. These somatic mutations were found to vary according to the subtype of ALL [13,14] and racial/ethnic disparity [15]. Recently, drug resistance-related mutations, such as those in *NT5C2*, *PRPS1*, *NR3C1*, *CREBBP*, and *TP53*, which facilitated relapse of pediatric ALL, were reported [16].

There are few reports analyzing the comprehensive and detailed mutation data of Korean pediatric patients with ALL using NGS. In this study, we aimed to investigate the mutational spectrum and frequency of pediatric ALL using a panel-based NGS with 67 leukemia-related genes at a single Korean institution and to reveal its clinical impact on disease relapse and survival outcomes.

## 2. Materials and Methods

### 2.1. Study Cohort

A total of 139 pediatric patients (≤18 years) who were diagnosed with either BCP-ALL (*n* = 123) or T-ALL (*n* = 16) and treated at our institution were enrolled in this study. The patients’ samples and clinical information between April 2018 and October 2020 were collected. We excluded cases of Philadelphia chromosome-positive ALL and Down syndrome and included cases of infant ALL (*n* = 6) in the study cohort. A diagnosis of ALL was confirmed via bone marrow (BM) pathology, immunophenotyping, cytogenetics, and molecular genetics. The presence of recurrent genetic abnormalities, including *ETV6-RUNX1*, *TCF3-PBX1*, *KMT2A*-rearrangement, high hyperdiploidy, and hypodiploidy, was diagnosed using reverse transcription polymerase chain reaction (RT-PCR), fluorescence in situ hybridization (FISH), and Giemsa band karyotyping. The cytogenetic and molecular genetic abnormalities were analyzed using BM samples at the time of diagnosis. All patients were risk-classified and treated according to an institutional protocol, which has been reported previously [17]. Clinical characteristics including age at diagnosis and initial white blood count (WBC), National Cancer Institute/Rome (NCI) risk group [18], as well as institutional risk group [17], were reported. The study received institutional review board approval from Seoul St. Mary’s Hospital, The Catholic University of Korea (IRB No. KC21RISI0481).

### 2.2. Conventional Cytogenetics and Fluorescence In Situ Hybridization

Chromosomal analyses were performed by examining short-term cultures of BM specimens according to standard conventional cytogenetic protocols. At least 20 cells in metaphase were analyzed in each case. Clonal abnormalities were classified according to the 2020 International System for Human Cytogenetic Nomenclature guidelines [19].

FISH analyses were performed to identify gene fusions, using a *TEL/AML1 (ETV6/RUNX1)* translocation dual-fusion probe (Cytocell, Banbury, UK), an *E2A (TCF3)* break-apart probe (Cytocell, Banbury, UK), and an *MLL (KMT2A)* break-apart probe (Cytocell, Banbury, UK). These FISH analyses used pellets of cells remaining after conventional cytogenetic studies. Slides for FISH were prepared by using cells harvested for conventional cytogenetics and by processing them for FISH according to the manufacturer’s guidelines (Abbott Vysis, Des Plaines, IL, USA). Analyses were performed on cells in either interphase or metaphase. 

### 2.3. Detection of Mutations Using Next-Generation Sequencing 

NGS was performed using a St. Mary’s customized NGS panel for acute leukemia, i.e., the “SM Acute leukemia panel”. Ion AmpliSeq Technology was used to amplify 67 genes (Appendix A) using an Ion Chef™ system and an Ion S5 XL Sequencer (all from Thermo Fisher Scientific, Waltham, MA, USA) [20]. Sequenced reads were mapped to the human reference genome (hg19, Genome Reference Consortium, February 2009). Annotated variants were classified into four tiers according to the Standards and Guidelines of the Association for Molecular Pathology [21]. Bioinformatics analysis was carried out using both customized and manufacturer-provided pipelines. Variants were selected and annotated using analytics algorithms and public databases [22]. All mutations were manually verified using the Integrative Genomic Viewer [23]. Among the 139 patients, 106 patients and 19 patients were analyzed with samples at diagnosis and at relapse, respectively, while 14 patients were analyzed with both samples. 

### 2.4. Multiplex Ligation-Dependent Probe Amplification

The targeted copy number of Ikaros family zinc finger 1 (*IKZF1*) was analyzed using a SALSA MLPA Probemix P335-C1 ALL-IKZF1 kit (MRC-Holland, Amsterdam, The Netherlands) according to the manufacturer’s instructions. The MLPA results were analyzed using GeneMarker software (Softgenetics, State College, PA, USA). Peak heights were normalized, and deletions or duplications were defined as recommended by the manufacturer. Direct sequencing of the probe-binding and ligation sites was performed for the relevant exons to detect nearby variants, which can lead to a false decrease in peak signal [24]. 

### 2.5. Statistical Analysis

A complex karyotype is defined as more than or equal to three chromosomal aberrations, including at least one structural aberration. Clinical characteristics, including median age and WBC count at diagnosis, and NCI and overall risk groups between non-relapsed and relapsed cases of ALL were compared using the Mann–Whitney test and Pearson’s chi-square test. The spectrum and number of genetic mutations for BCP-ALL and T-ALL were compared using Fisher’s exact test and the Kruskal–Wallis test. Event-free survival (EFS) and OS were determined via the Kaplan–Meier method, with comparisons of survival rate performed using a log-lank test. The date of the last follow-up was 28 February 2021. *p* < 0.05 was considered statistically significant. SPSS version 24.0 (SPSS, Chicago, IL, USA) was used for all statistical analyses.

## 3. Results

### 3.1. Genetic Characteristics of the Study Cohort

Among 139 pediatric patients with ALL, seventy-eight (56.1%) patients had recurrent genetic abnormalities (ETV6-RUNX1 in 34, high hyperdiploidy in 29, TCF3-PBX1 in 7, KMT2A-rearrangement in 7, and hypodiploidy in 1), and nine (6.5%) patients had complex karyotypes. When comparing the two phenotypes, recurrent genetic abnormalities were more frequently found in BCP-ALL (*p* < 0.001), whereas a complex karyotype was more dominant in T-ALL (*p* = 0.001). Eighty-five patients (61.2%) harbored at least one genetic mutation that seemed either pathogenic (P) or likely pathogenic (LP), and the median number of these mutations was 1 (range, 1–10). The number of mutations in patients with BCP-ALL was significantly lower than that in patients with T-ALL (0.86 ± 1.081 vs. 2.75 ± 2.206) (*p* < 0.001). The main pathways detected via the NGS panel of this study were RAS and NOTCH signaling pathways and transcriptional regulation. The five most frequently mutated genes in the total cohort were NRAS, KRAS, NOTCH1, FBXW7, and PTPN11, in that order (Figure 1). We also analyzed the co-occurrence of mutations and complex karyotypes. Several pairs of co-occurring mutations were found, such as NRAS with SETD, NRAS with PTPN11, and NOTCH1 with FBXW7 (*p* = 0.024, *p* = 0.020, and *p* < 0.001, respectively) (Figure 2) [25]. 

### 3.2. Mutational Spectrum of Pediatric ALL According to Disease Category

#### 3.2.1. BCP-ALL: Mutations in the RAS Pathway Were More Abundant

In BCP-ALL (*n* = 123), 68 patients (55.3%) harbored mutations, and the median number of mutations was 1 (range, 0–3). The five most frequently mutated genes in BCP-ALL were NRAS (22.4%), KRAS (19.6%), PTPN11 (8.4%), TP53 (8.4%), and FLT3 (7.4%), in that order (Figure 3A). The number of somatic mutations in patient groups with ETV6-RUNX1 (0.38 ± 0.70) was significantly lower than that in patient groups with no recurrent genetic abnormality (1.11 ± 1.18), high hyperdiploidy (1.17 ± 0.90), and KMT2A rearrangement (1.17 ± 0.98) (*p* < 0.001, *p* < 0.001, and *p* = 0.037, respectively). A total of 47 (38.2%) out of 123 patients showed mutations in the RAS pathway (NRAS, KRAS, PTPN11, FLT3, and NF1 in order), which is the most commonly involved pathway in BCP-ALL. All RAS pathway mutations found in this study were missense mutations (Figure 1). RAS pathway mutation had a particularly high frequency in patients with high hyperdiploidy (*n* = 34); 19 patients (67.8%) with high hyperdiploidy carried the RAS pathway mutation. NRAS and PTPN11 were significantly associated with a high hyperdiploidy karyotype (*p* = 0.018 and *p* < 0.001, respectively). IKZF1 deletions were detected in 10 patients (8.1%) with BCP-ALL, and three and seven patients showed whole and partial gene deletions, respectively. The two main detected focal deletions were exons 4–7 and exons 4–8.

#### 3.2.2. T-ALL: Mutation in the Notch Pathway Were More Common

All patients with T-ALL (*n* = 16) harbored at least one mutation and the median number of mutations was 3 (range, 1–10), which was significantly higher than that in BCP-ALL (*p* < 0.01). The three most frequently mutated genes in T-ALL were NOTCH1 (37.5%), FBXW7 (16.6%), and PTEN (6.2%), in that order. In particular, NOTCH1, FBXW7, PTEN, and WT1 mutations were found more frequently in T-ALL than in BCP-ALL (*p* < 0.001, *p* < 0.001, *p* = 0.013, and *p* = 0.013, respectively) (Figure 3B). Among T-ALL patients, a complex karyotype was significantly associated with NOTCH1, PTPN11, and FBXW7 mutations (*p* < 0.001, *p* = 0.048, and *p* = 0.006, respectively) (Figure 2). Five (31.3%) patients harbored both NOTCH1 and FBXW7 mutations, a significant co-occurring mutation in T-ALL. 

#### 3.2.3. Relapsed ALL: IKZF1 Deletion and NT5C2 Mutation Seemed to Associate with Disease Relapse

The changes in mutations between the time of diagnosis and the time of relapse are shown in Figure 4. A higher number of mutations was observed in patients who experienced relapse (*n* = 34) than in patients who did not experience relapse (*p* < 0.01). We compared the mutation changes in 14 patients for whom both diagnostic and relapse samples were available: eight patients showed the same mutations, five patients had newly emerged mutations, and one patient lost the initial mutation. The most frequent newly emerged mutation was NT5C2 (42%), which was found in three of three cases of relapse. In addition to the NT5C2 gene, ETV6, SETD2, and IL7R were newly observed genes in relapsed samples. Two of two patients with PTEN and CDKN2A, four out of eight patients with TP53, and three out of six patients with CREBBP relapsed, respectively. There was no difference in clinical outcomes according to presence of TP53 mutation in our cohort. The proportion of patients who showed IKZF1 deletion was higher in the relapsed ALL group (17.6%) than in the non-relapsed ALL group (3.8%) (*p* < 0.001) (Table 1).

#### 3.2.4. Comparison of Clinical Characteristics Based on Relapse

The median age at diagnosis was 5.0 years (range, 0.1–18.0) in the study cohort. In T-ALL cases, the median age at diagnosis and the median value of initial WBC counts were significantly higher than those in BCP-ALL cases (*p* < 0.01). When comparing the clinical characteristics between non-relapsed and relapsed cases of ALL, we found that the median value of initial WBC count was significantly higher in relapsed ALL (25.5 × 10^9^/L, range 0.9–630) than in non-relapsed ALL (10.5 × 10^9^/L, range 0.6–339) (*p* < 0.001). In addition, more patients in the NCI high-risk (HR) group experienced relapse than those in the NCI standard-risk (SR) group (NCI HR 67.7% vs. NCI SR 40.9%, *p* = 0.016). 

The 3-year EFS and OS of the total cohort were 65.8 ± 8.5% and 87.1 ± 5.3%, respectively. There were no significant differences in survival outcomes between BCP-ALL and T-ALL (EFS: BCP-ALL 66.5% ± 8.6% vs. T-ALL 80.8 ± 10.0%, *p* = 0.881; OS: BCP-ALL 89.0 ± 5.3% vs. T-ALL 79.1% ± 10.8%, *p* = 0.160). When compared with patients in the NCI SR group, patients in the NCI HR group showed an inferior EFS (NCI HR 62.2 ± 8.8% vs. NCI SR 66.8 ± 14.7%, *p* = 0.011) and comparable OS (NCI HR 87.0 ± 5.2% vs. NCI SR 85.0 ± 10.7%, *p* = 0.082). The 3-year OS of patients who experienced relapse was 76.5 ± 7.8%, while no patient died of disease among those who did not experience relapse. No significant differences were found in the EFS and OS according to the response in the corticosteroid window phase, institution risk group, presence of RAS pathway mutation, IKZF1 deletion, and presence of recurrent genetic abnormalities. 

## 4. Discussion

To assess the mutational spectrum in Korean pediatric ALL, we performed a panel based NGS analysis that involved various signaling pathways and showed comprehensive genetic profiles among these patients. This study showed a distinctive mutational spectrum and frequency between BCP-ALL and T-ALL; in BCP-ALL, the dominant mutations were enriched in the RAS pathway, while in T-ALL, the dominant ones were those in the NOTCH pathway. Though no significant mutations predicting disease prognosis were found in our cohort, *IKZF1* deletion and *NT5C2* mutation seemed to be associated with disease relapse.

As previous studies reported, RAS pathway mutations were well-known recurrent somatic variants in pediatric BCP-ALL, and the vast majority of mutations occurred in *KRAS*, *NRAS*, *PTPN11*, *FLT3*, and *NF1* [26,27]. We reported similar results that the most common somatic mutations were in the RAS signaling pathway, and all these mutations were missense mutations. RAS pathway mutations have been associated with disease relapse [28] and chemotherapy resistance [29] in pediatric BCP-ALL. However, the prognostic significance of clonal and subclonal mutations remains unknown. Although RAS pathway mutations were not correlated with relapse in our cohort, all RAS pathway mutations observed at the time of relapse were clonal mutations, except in one patient. This finding deserves further analysis with a larger cohort in future studies.

Although ALL with high hyperdiploidy (51–67 chromosomes) generally represents a favorable prognostic subtype, less than 20% of these patients experienced disease relapse [30]. Studies have reported a high incidence of RAS pathway mutations in patients with high hyperdiploidy and have linked it to disease recurrence [27,31]. We also observed that RAS pathway mutations occurred in more than half of ALL patients with high hyperdiploidy (67.8%) with a significant association, particularly between a high hyperdiploidy karyotype and *NRAS*-*PTPN11* mutation. In this study, ALL patients with high hyperdiploidy harbored more somatic mutations than patients with other recurrent translocations, such as *ETV6-RUNX1* and *TCF3-PBX1*. Among them, *ETV6-RUNX1*-positive ALL patients harbored the least number of somatic variants.

A deletion of *IKZF1* is known to be associated with adverse outcomes and has been described as a high-risk marker in pediatric BCP-ALL [32]. Most of the published data on *IKZF1* deletions in pediatric ALL have been generated via MLPA analysis and have revealed its reliability to detect all deletions targeting the coding sequence [33,34]. We reported a *IKZF1* deletion frequency of 8.1%, which is slightly lower than previous findings of an overall frequency around 15% [35]. The presence of *IKZF1* deletions was linked to features indicating poor prognosis, including an older age at diagnosis, higher initial WBC count, and higher levels of minimal residual disease after induction chemotherapy [34,35]. As expected, 6 out of 10 patients with *IKZF1* deletion were risk-stratified into a ‘very high risk’ group because of either high WBC count or older age at presentation, and significantly more patients with *IKZF1* deletion experienced disease relapse compared to those with wild-type *IKZF1* in this study. These findings reiterate the urgency of introducing new specific therapeutic approaches for this patient group.

Recently, several studies on pediatric ALL reported that mutations in *NT5C2* caused disease relapse, especially during maintenance chemotherapy, by conferring purine analog resistance [36,37,38]. In this study, *NT5C2* mutations were found to have newly emerged in three patients with relapsed ALL. Among them, two patients with subclonal mutation died after relapse while one patient with clonal mutation remained alive throughout the study period. Barz et al. reported that subclonal *NT5C2* mutations independently predict an inferior outcome of disease relapse [37]. In addition to alterations in *NT5C2*, relapse-specific somatic alterations, such as those in *TP53* and *CREBBP*, were also more frequently found in patients who experienced relapse in our cohort.

Nevertheless, this study has some limitations. Primarily, the mutational analysis was performed in a disease-only setting. We could not obtain germline samples, such as skin fibroblasts, and did not perform a familial study to search for germline mutations. In addition, samples from the time of diagnosis were not available for all patients who experienced relapse; therefore, the mutational difference between the time of diagnosis and relapse could not be fully elucidated. Third, we used a custom panel for NGS that only including 67 leukemia-related genes. The use of a wider panel (e.g., commonly used 409-gene panel) in future study could provide more comprehensive results. Finally, although our study employed one of the largest cohorts of Korean pediatric patients, a relatively smaller number of cases were enrolled than in multicenter studies.

## 5. Conclusions

We found a difference in recurrent somatic mutations between BCP-ALL and T-ALL in Korean pediatric patients. Our findings strengthen the current knowledge of genetic profiles in pediatric ALL. Furthermore, we demonstrate the usefulness of panel-based NGS techniques in providing comprehensive genetic information. We expect that these data will serve as a foundation to develop a mutational roadmap for Korean pediatric patients for nation-wide studies in the future.

## Figures and Tables

**Figure 1 jcm-11-06298-f001:**
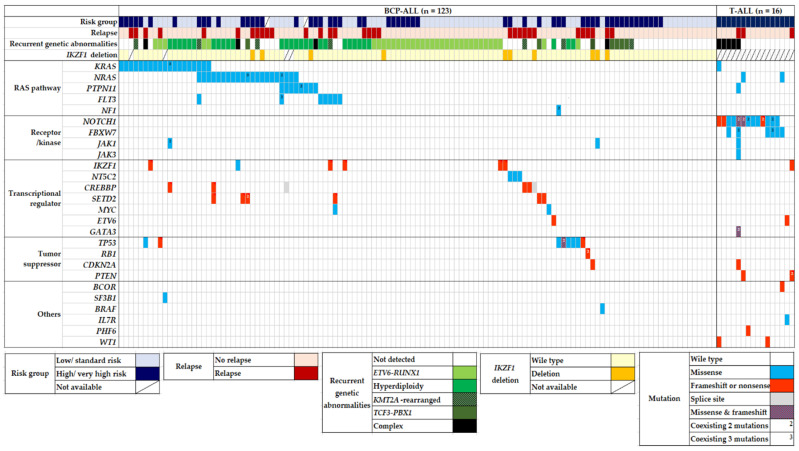
Mutational landscape of 139 pediatric ALL patients. Heatmap diagram showing genomic data of recurrent translocations, *IKZF1* deletion, and somatic mutations in pediatric ALL. Abbreviation: BCP-ALL, B-cell precursor acute lymphoblastic leukemia.

**Figure 2 jcm-11-06298-f002:**
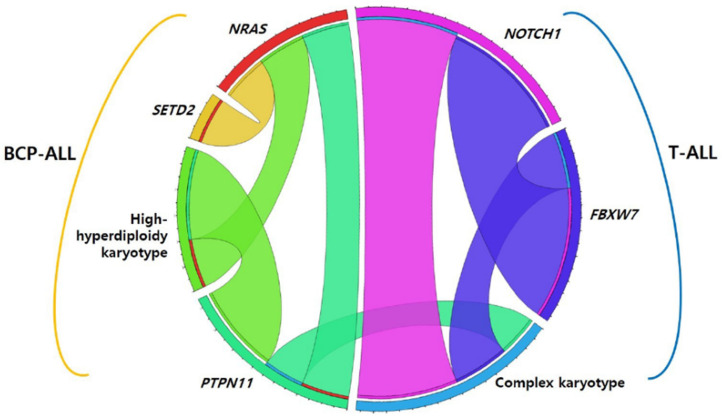
Co-occurring mutations in pediatric acute lymphoblastic leukemia. Abbreviation: BCP-ALL, B-cell precursor acute lymphoblastic leukemia.

**Figure 3 jcm-11-06298-f003:**
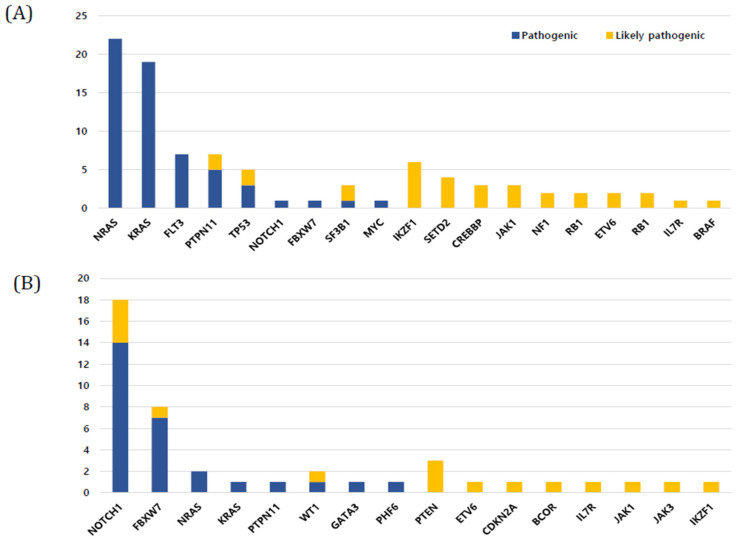
Frequency of mutations in study cohort. (**A**) B-cell precursor acute lymphoblastic leukemia (ALL) (**B**) T-cell ALL.

**Figure 4 jcm-11-06298-f004:**
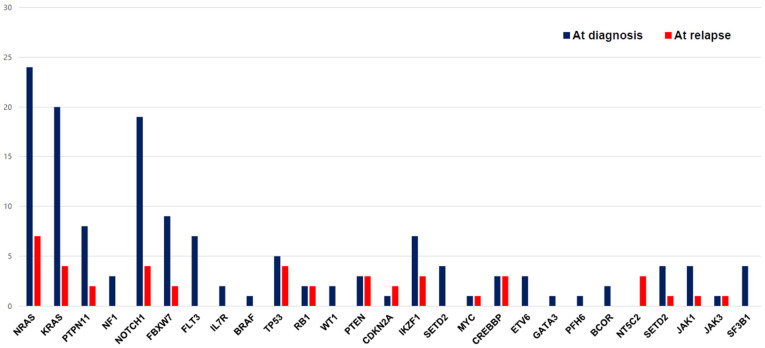
The change in mutational distribution of the study cohort at the time of diagnosis and relapse.

**Table 1 jcm-11-06298-t001:** Comparison of clinical characteristics between non-relapsed and relapsed ALL.

	Non-Relapsed ALL (*n* = 105)	Relapsed ALL (*n* = 34)	
Median age at diagnosis (range)	5.0 (0.5–17.0)	9.5 (0.1–18.0)	0.176
Median WBC at diagnosis, 10^9^/L (range)	10.5 (0.6–339)	25.5 (0.9–630)	<0.001
NCI risk group, *n* (%)			
Standard risk	62 (59.1)	11 (32.3)	
High risk	43 (40.9)	23 (67.7)	0.016
Overall risk group, *n* (%)			
Low risk	28 (26.6)	3 (8.8)	
Standard risk	26 (24.7)	10 (26.4)	
High risk	22 (20.9)	7 (20.5)	
Very high risk	29 (27.6)	14 (38.2)	0.066
Steroid response, *n* (%)			
Good	91 (81.6)	19 (55.8)	
Poor	10 (9.5)	5 (14.7)	
Unknown	4 (3.9)	10 (29.5)	0.475
Recurrent genetic abnormalities, *n* (%)			
High hyperdiploidy	24 (22.8)	5 (14.7)	
*ETV6-RUNX1*	29 (27.6)	4 (11.7)	
*KMT2A*-rarrangement	3 (2.8)	3 (8.8)	
*TCF3-PBX1*	5 (4.7)	2 (5.8)	
Hypodiploidy	1 (0.9)	1 (2.9)	
Not detedted	43 (40.9)	19 (52.1)	0.210
*IKZF1* deletion, *n* (%)	4 (3.8)	6 (17.6)	<0.001
RAS pathway mutation	31 (29.5)	12 (35.2)	0.830

Abbreviation: ALL, acute lymphoblastic leukemia; WBC, white blood count; NCI, national cancer institute.

## Data Availability

Not applicable.

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
