# Peer review of "Spectrum of Genetic Mutations in Korean Pediatric Acute Lymphoblastic Leukemia"

_jcm, 2022, doi:10.3390/jcm11216298_

Round 1
Reviewer 1 Report
This is a quite interesting and well-written paper.
The Authors performed and extensive genetic characterization of pediatric ALL in their country.
I have only one comment. The Authors used a custom panel for NGS. A wider (ed the commonly used 409 genes panel) panel might have been of interest. The Authors have to first explain better the rational for 67 genes selection and then discuss the possible limitations
Author Response
Response to Reviewer 1 Comments
We would like to sincerely thank the reviewers for taking the time to read and critique our paper. We would like to respond to each comment in the following manner.
Point 1: The Authors used a custom panel for NGS. A wider (ed the commonly used 409 genes panel) panel might have been of interest. The Authors have to first explain better the rational for 67 genes selection and then discuss the possible limitations.
Response 1: We appreciate your comments. Our customized panel includes well-established & commonly mutated genes detected in pediatric ALL and was well validated for clinical use (Kim et al. Blood Cancer J 2021,11,109). However, we do agree with your insightful suggestion that a wider panel might be more interesting for research, so we added this limitation in the discussion (Revised manuscript page 9, line 294~296)
CLOSING COMMENTS TO THE EDITOR:
Again, we appreciate all your insightful comments. We worked hard to be responsive to them. Thank you for taking the time and energy to help us improve the manuscript.

Reviewer 2 Report
Dear Authors,
your manuscript decribe very important topic of mutations in childhood B- and T-ALL. However I have some questions:
1) It is well known that subpopulation of hematopoietic stem cells (HSC) that carry mutations responsible for the onset of leukaemia and its relaps are very hard to kill by the treatment. Could you monitor spectrum of mutations in HSCs.
2) Can some of the mutation serve as a biomarker for the early detection of leukaemia? What is the frequency of these mutations in healthy non-leukemic population?
3) Did you take the samples also in the state of remission? Do you expect leukemogenic mutation to be present during remission in relapsing vs. non-relapsing patients? Mutations detected in remission could probably be from the leukaemia - initiating stem cells that are dormant and hardly targeted by chemotherapy, because the same mutations were present in relapse as compared to diagnosis in some cases?
4) How can your data be uses in clinical setting in future?
Author Response
Response to Reviewer 2 Comments
We would like to sincerely thank the reviewers for taking the time to read and critique our paper. We would like to respond to each comment in the following manner.
Point 1: It is well known that subpopulation of hematopoietic stem cells (HSC) that carry mutations responsible for the onset of leukaemia and its relaps are very hard to kill by the treatment. Could you monitor spectrum of mutations in HSCs.
Response 1: We appreciate your comments. Our B-ALL monitoring strategy is flow cytometry, conventional cytogenetics, FISH, and Ig gene rearrangement NGS, so unfortunately there is no mutation monitoring data. Recently, we demonstrated that Ig gene rearrangement NGS MRD is a significant prognostic indicator (Front Oncol. 2022 Sep 15;12:957743. doi: 10.3389/fonc.2022.957743). However, as you pointed out, it is also necessary to study about meaningful mutations responsible for the onset of leukaemia and its relapse in the future.
Point 2: Can some of the mutation serve as a biomarker for the early detection of leukaemia? What is the frequency of these mutations in healthy non-leukemic population?
Response 2: Primary abnormalities cause the formation of a pre-leukemic clone, which then develops into overt leukaemia with the addition of secondary or cooperating genetic changes. Primary abnormalities in ALL are often chromosomal abnormalities, or aneuploidy; whereas secondary abnormalities are usually genetic mutations. The RAS pathway and TP53 mutations have been recognized as important additional drivers of these unique ploidy subgroups in recent research (Nat Genet. 2013;45(3):242-252). KRAS mutation is found in 0.8% of healthy non-leukemic population, according to a meta-analysis of studies (Biomark Med. 2009;3(6):757-69).
Point 3: Did you take the samples also in the state of remission? Do you expect leukemogenic mutation to be present during remission in relapsing vs. non-relapsing patients? Mutations detected in remission could probably be from the leukaemia - initiating stem cells that are dormant and hardly targeted by chemotherapy, because the same mutations were present in relapse as compared to diagnosis in some cases?
Response 3: Thank you for reviewer’s insightful comments. Although we didn’t analyze the samples at the time of remission due to retrospective manner in this study, however, we are prospectively collecting patient’s samples at diagnosis, at remission, and at relapse. As per reviewer’s suggestion, we also have plan to analyze potential leukemogenic and/or drug resistant mutation by comparing genetic mutations in each time point in pediatric ALL.
Point 4: How can your data be uses in clinical setting in future?
Response 4: We appreciate your comments. In our thought, and as you suggested, one of limitation of our study is that we did not have all the samples at all time points (e.g., at diagnosis, at remission, and at relapse). If we can identify some genetic mutations which were frequently detected in relapse sample, it is expected that we can suggest more detailed risk stratification and treatment modification for these patients.
CLOSING COMMENTS TO THE EDITOR:
Again, we appreciate all your insightful comments. We worked hard to be responsive to them. Thank you for taking the time and energy to help us improve the manuscript.

Reviewer 3 Report
Yoo et al. discuss the genetic mutations in Korean Pediatric ALL. The manuscript is mainly descriptive and discusses some mutations in detail but not much about TP53. I would recommend discussing in short TP53 results and how they may affect the outcomes. Although the number is not large, we know TP53 is one of the mutations with the worst outcome in any time of cancer and would be interesting to know its effects on ALL.
Author Response
Response to Reviewer 3 Comments
We would like to sincerely thank the reviewers for taking the time to read and critique our paper. We would like to respond to each comment in the following manner.
Point 1: Yoo et al. discuss the genetic mutations in Korean Pediatric ALL. The manuscript is mainly descriptive and discusses some mutations in detail but not much about TP53. I would recommend discussing in short TP53 results and how they may affect the outcomes. Although the number is not large, we know TP53 is one of the mutations with the worst outcome in any time of cancer and would be interesting to know its effects on ALL.
Response 1: We appreciate your comments. In our cohort, eight patients had TP53 mutation and 4 out of 8 patients experienced disease relapse. We analyzed the clinical outcomes between TP53 mutated and TP53 wild-type patients, and we found that there were no differences in clinical outcome between two groups. However, we agree with reviewer’s comment regarding TP53 mutation may affect the outcomes. As per reviewer’s suggestion, we added the outcome of patients with TP53 mutation in the result session. (Revised manuscript page 5, line 194~195)
CLOSING COMMENTS TO THE EDITOR:
Again, we appreciate all your insightful comments. We worked hard to be responsive to them. Thank you for taking the time and energy to help us improve the manuscript.